# Quantitative Ultrasound Analysis of Oral Mucosa: An Observational Cross-Sectional Study

**Dario Di Stasio** [1], **Antonio Romano** [1], **Marco Montella** [2], **Maria Contaldo** [1], **Massimo Petruzzi** [3], **Iquebal Hasan** [4], **Rosario Serpico** [1] and **Alberta Lucchese** [1,*]

1     Multidisciplinary Department of Medical-Surgical and Dental Specialties, University of Campania "Luigi Vanvitelli", 80138 Naples, Italy; dario.distasio@unicampania.it (D.D.S.); antonio.romano4@unicampania.it (A.R.); maria.contaldo@unicampania.it (M.C.); rosario.serpico@unicampania.it (R.S.)

2     Department of Advanced Medical and Surgical Sciences, University of Campania "Luigi Vanvitelli", 80138 Naples, Italy; marco.montella@unicampania.it

3     Interdisciplinary Department of Medicine, University of Bari, 70121 Bari, Italy; massimo.petruzzi@uniba.it

4     UNC Adams School of Dentistry, 385 S Columbia St., Chapel Hill, NC 27599, USA; hasaniq@unc.edu

\*     Correspondence: alberta.lucchese@unicampania.it; Tel.: +39-081-566-7670

**Featured Application: The quantitative analysis of ultrasound images of the oral mucosa could allow us to standardize the use of ultrasound in the oral mucosa, limiting operator-dependent variability and allowing the creation of computer-assisted models.**

**Abstract:** (1) Background: Ultrasonography is gaining popularity as a diagnostic tool in the study of the oral mucosa. The precision of ultrasound has made it possible to identify the various layers, based on their echogenicity. The aim of this study was to perform a quantitative analysis of healthy oral mucosa based on the analysis of greyscale, echo levels (dB), and attenuation values (dB/cm). (2) Methods: Thirty-three patients (17 females and 16 males; $58.42 \pm 13.29$ y.o) were recruited for this study. The images were acquired with the GE Logiq-e R7 with a linear probe at 18 MHz frequency (harmonic). For each tissue (epithelium, rete ridges, connective tissue, muscle, and bone), regions of interest were traced for the analysis of echo levels, grey levels, and attenuation values. One-way ANOVA and pairwise comparison were performed. (3) Results: Three-hundred and thirty images were analyzed. Analysis of echo levels and grey levels showed a significant difference between epithelium and rete ridges ($p = 0.001$), and between rete ridges and connective tissue ($p = 0.001$), but not between epithelium and connective tissue ($p = 0.831$) or connective and muscle layers ($p = 0.383$). The attenuation values appeared to be specific for each tissue layer ($p = 0.001$). (4) Conclusions: Quantitative analysis applied to ultrasound imaging of the oral mucosa allows the definition of specific tissue areas.

**Keywords:** ultrasound imaging; oral mucosa; quantitative analysis

## 1. Introduction

Ultrasonography (US) is a diagnostic medical tool that uses high-frequency sound waves and their echoes to create images of the internal body structures with frequencies between 1.6 and 50 MHz [1,2]. Ultrasound imaging is based on the waves echoing off the body structures being examined, resulting in different grades of grey, from white to black [2]. The shades are based on tissue density: the denser the tissue is, the lighter it will be. Bone appears bright white, and fluid appears black. Wave reflection or transmission depends on the acoustic impedance of the different tissues: this varies according to the density of the material ultrasound passes through. It is the product of the tissue's density and the sound's velocity through it [3]. Other non-invasive imaging techniques have been used in the oral cavity to study the anatomical and pathological characteristics of the oral mucosa [4–6].

Noninvasive imaging procedures performable chairside and that show more information in real-time without harming the patient could remarkably enhance the quality and the speed of diagnosis and enable the patient's compliance [7]. These include optical coherence tomography (OCT—which transforms the infrared backscatter into grayscale), reflectance confocal microscopy (RCM—which allows analysis at the cellular and subcellular levels of hard and soft oral tissues), and virtual chromoendoscopic magnification and narrow-band imaging (NBI) [8–11].

The properties of ultrasound have enabled us to obtain different images, of which the most common are B-mode (brightness mode) and A-mode (amplitude mode). A-mode represents the amplitude of reflected ultrasound plotted against the transit time or, if the speed of sound in the tissue is known, against the transit distance. Therefore, A-mode sonography is increasingly being replaced by B-mode sonography, which provides a tomographic image by representing the amplitude of reflected sound linearly encoded in shades of gray as a function of axial distance from the sound source [12]. The echo returning to the transducer is converted to dots. Their width corresponds to the echo at its base and the brightness to the received amount of energy. Thus, the more the waves were reflected, the brighter the spot. Each point is bright (hyperechoic) or dark (hypoechoic), or completely black (anechoic), corresponding to the intensity of echoes from the corresponding anatomic structure. In this way, a B-mode scan provides an image that resembles an anatomical cross-section of the tissue being scanned [13].

In head and neck surgery, ultrasonography has been used to diagnose neck masses, salivary gland tumors, tongue carcinoma, and other oral lesions. In examining oral lesions, ultrasonography was carried out by placing a transducer on an extraoral site, but it was difficult to obtain detailed data.

An extraoral approach has some limitations: it does not provide satisfactory images, particularly of the tongue, palate region, and oral floor, because air spaces within the oral cavity attenuate acoustic waves, and ultrasound does not penetrate the bone well. Furthermore, the acoustic attenuation generated by the distance from the lesion to the skin could become an issue [14]. Nevertheless, the extraoral approach is crucial for some applications, such as temporomandibular joint disorders [15].

The intraoral ultrasonographic examination is non-invasive, rapid, and easily repeated; the number of possible applications has thus increased [16]. This technique has been used in the oral cavity for various purposes: from the study of periodontal tissues to the clinical and presurgical characterization of benign and malignant oral mucosa lesions [17–21]. One of the possible usages of ultrasonography is exposing an impacted tooth/teeth, and to establish if the tooth is covered by bone or not, as this can determine whether one can use less invasive methods, such as a diode laser or the conventional scalpel exposure [22].

According to Chan et al., the two methods are superimposable and must be selected based on the anatomical position of the area to scan with ultrasound. It is recommended to execute both approaches in some regions to have full details [1].

Various authors deciphered the sonographic characteristics of healthy oral mucosa [6]. With the progress of technology, some recent ex vivo and in vivo studies suggested that US can discern tissue layers based on greyscale images through acoustic backscatter [12].

The primary consideration in assessing the accuracy of an imaging technique is the grade to which that procedure matches directly with histopathology [20].

Quantitative analysis of echograms for medical diagnosis improvement (support) has received broad interest over the past 30 years [23,24]. By analyzing the texture of echographic images, additional information is gathered on the tissue type.

The technique is based on the production of high-frequency sound waves directed at the tissues. The interaction between the waves and the tissues will produce return waves that will be picked up by the same emission source and processed in grayscale images [3]. A more reflective tissue will return to the source an ultrasound beam of very similar intensity compared to the original one, whereas a less compact one will tend to be more easily crossed by the beam [13]. The value of the returned beam corresponds to echo intensity

(echo levels—ELs), and each value, or range of values, of EL will correspond a gray value from 0 to 255. EL is measured in decibels (dB) and is linearly related to the intensity. Zero dB equals max intensity, i.e., white with gray level 255 [25].

This observational cross-sectional study aimed to:

- obtain multiple, high-definition US images in different regions of the healthy oral cavity.
- perform a quantitative analysis of the high-definition US images obtained based on the analysis of greyscale (GL), echo levels (ELs—dB), and attenuation values (ATT—dB/cm),

with the final goals of defining the oral mucosa's ultrasound characteristics and making it possible to identify the various layers based on their echogenicity.

## 2. Materials and Methods

Appropriate ethical approval was secured by the Ethics Committee of the University of Campania "L. Vanvitelli" (number 339/2018). All procedures performed in studies involving human participants were in accordance with the ethical standards of the institutional and/or national research committee and with the 1964 Helsinki declaration and its later amendments or comparable ethical standards.

Patients were recruited consecutively at the Oral Pathology Unit at the University of Campania L. Vanvitelli, Naples, Italy, from December 2021 to March 2022, after giving written informed consent. Patients over 18 with at least one or more sites of healthy oral mucosa were included in the study. The following lesions and or conditions were considered as exclusion criteria: (a) potentially malignant oral disorders; (b) oral squamous cell carcinoma; oral pigmented lesions; (c) erosive-erythematous lesions of inflammatory and/or autoimmune etiology; (d) severe broken-down teeth; (e) cleft lip; (f) palate lip; (g) orthodontic treatments; (h) smoking and alcohol consumption; (i) pregnant patients. The US images' real-time acquisition and interpretation were performed at the end of the clinical examination. Ultrasound images of healthy oral mucosa were collected.

### 2.1. Ultrasound Investigations

The images were acquired with the GE Logiq-e R7 (General Electric, GE Healthcare division, Milan, Italy) device with a 15″ LCD monitor, with a resolution of 1024 × 768. The transducer used was a GE L8-18i-D "hockey-stick" linear probe, with a central frequency range of 5–18 MHz, an axial field of view from 0.4 mm to 2 cm, and a side field of view from 0.7 mm to 2 cm. Images were acquired in B-mode at the maximum transmit frequency (18 MHz, harmonic), and a gain set to 45 and 60 dB of dynamic range. The examination was performed after applying ultrasound gel to the scanning area. A fixed distance of about 5 mm between the area to scan and the probe was set. The focus was standardized [20].

### 2.2. Study Sample

A total of 33 patients (17 females and 16 males; mean age 58.42 ± 13.29 y.o; age range 29–80 years) were considered for this study. The full entire series encloses a total of 1002 ultra-sound images of oral cavities covering all the anatomy sites: 334 images of masticatory mucosa (MM), 328 images of buccal mucosa (BM), and 340 images of the tongue. Ten images that met the quality standards were selected for each patient, for a total of 330 images of healthy mucosa. In each image, it was possible to identify all the tissue layers based on the anatomical site.

### 2.3. Dataset

All the images were acquired by a single operator with five years of experience with the technique and the equipment. Ten images of satisfactory quality and homogeneous characteristics for each mucosal site were selected for analysis by two operators. All images were cataloged using Horos imaging software (version 3.3.6—Horos Project, Annapolis, MD, USA) [26]. All data were anonymized. In the post-acquisition phase, a specific region of interest (ROI) for each image was manually defined in each tissue layer, by two

independent operators, through an embed function within the operating system of the ultrasound device. This function allows operators to measure the average echo levels (ELs) in decibels (dB) within the ROI. The diameter of the ROI was variable according to the thickness of the selected tissue, from a minimum of 100 μm to a maximum of 500 μm. The tissues analyzed through the ROIs were epithelium (EP), rete ridges (RR), connective tissue (CL), the muscular layer (ML) for the BM, and the tongue and bone for the MM (Figure 1). The scanned areas are reported in Table 1. The ELs were subsequently converted to 8-bit grayscale (GL). Zero dB equals max intensity, i.e., white with gray level 255. Minimum intensity equals –99 dB, i.e., black with gray level zero. The intraclass correlation coefficient (ICC) was calculated to assess inter-rater reliability.

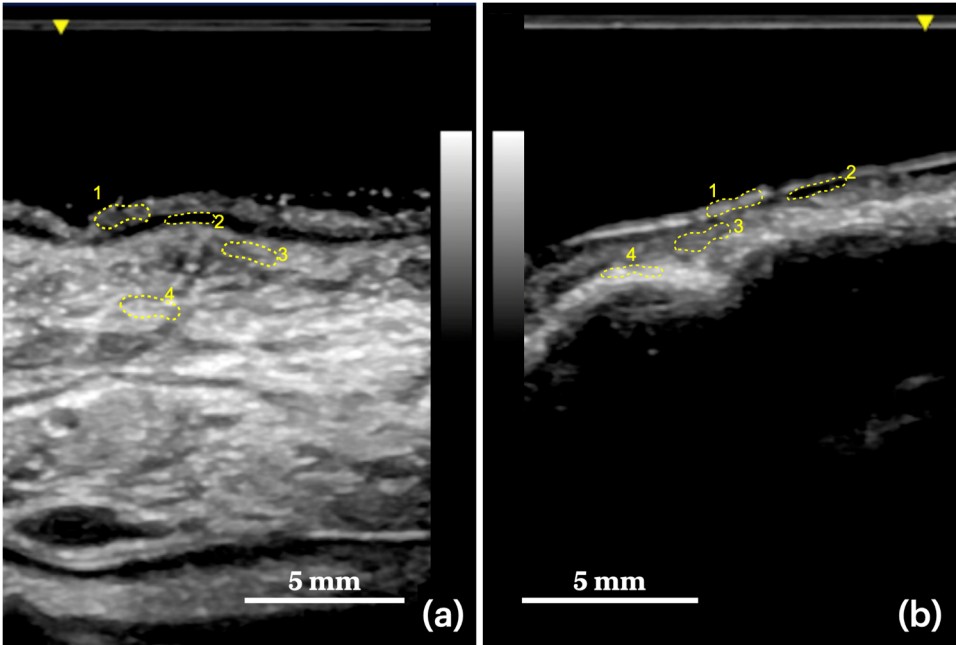

**Figure 1.** Two examples of ROI sampling in two anatomical regions of the oral cavity. (**a**) Dorsal tongue: (1) epithelium, (2) rete ridges, (3) connective layer, (4) muscle layer. (**b**) Ultrasound imaging of the upper gingiva and alveolar bone (masticatory mucosa): (1) epithelium, (2) rete ridges, (3) connective layer, (4) alveolar bone.

**Table 1.** Regions and specific areas of the oral cavity scanned with ultrasonography.

| | |
|---|---|
| **Masticatory mucosa** | upper gingiva (vestibular side)<br>hard palate (palatine crests)<br>maxillary alveolar process<br>lower gingiva (buccal and lingual side) |
| **Buccal mucosa** | cheeks mucosa<br>upper and lower lips (mucous side) |
| **Tongue** | dorsal tongue<br>lateral margins<br>ventral tongue<br>floor of the mouth |

### 2.4. Image Processing

One of the main problems with the quantitative analysis of ultrasound images is that the impedance, and consequently, the gray levels, can depend on the depth of the tissue [24]. This was compensated for by standardizing the distance between the area to be scanned and the probe. In addition, a variable was created with the attenuation values (ATT) of each scanned ROI in relation to the distance from the probe (dB/cm) [24,27].

### 2.5. Data Analysis and Statistics

Descriptive statistical analyses of quantitative ultrasound parameters (mean, standard deviation (SD) minimum, and maximum), the mean distance of the ROI from the transducer, and patient demographics were performed. The mean values of the ultrasound parameters (EL, GL, ATT) were compared in the five subgroups (EP, RR, CL, ML, and bone) through one-way ANOVA and post hoc pairwise comparison. The significance level was set to *0.05*. Bonferroni correction was applied for multiple comparisons, and *p = 0.01* was established for statistical significance. Analyses were performed using IBM SPSS (IBM SPSS Statistics for Mac version 28.0.1.0—IBM Corp., Armonk, NY, USA). The graphs were made with GraphPad Prism Version 9.3.1 (GraphPad Software, San Diego, CA, USA).

To assess the statistical power and significance of CSI variation in both groups, a power analysis was performed using G*Power 3.1.9.6 (Heinrich Heine University, Düsseldorf, Germany) for one-way ANOVA, assuming a medium Cohen's effect size of 0.5 ($\alpha = 0.05$). Sensibility (statistical power) was fixed at 0.80.

### 3. Results

This study analyzed 330 images of healthy oral mucosa. The mean scan distance was 4.378 ($\pm$0.86) mm, which corresponded to the EP's target depth (TD). The other TDs and the ICC are shown in Table 2.

**Table 2.** Descriptive statistics of quantitative ultrasound parameters.

| Tissue Layers | US Parameters | EL (dB) | GL (8-bit) | ATT (dB/cm) | TD (cm) | ICC (Sig.) |
|---|---|---|---|---|---|---|
| EP | *Mean* | −41.073 | 149.205 | −97.268 | 0.438 | |
| | *SD* | 5.84 | 15.04 | 25.21 | 0.09 | 0.871 (*p = 0.001*) ** |
| | *Minimum* | −50.44 | 125.08 | −143.20 | 0.27 | |
| | *Maximum* | −29.11 | 180.01 | −45.78 | 0.64 | |
| EP-RR | *Mean difference (SE)* | 26.18 (±1.77) | 73.21 (±4.34) | 49.73 (±7.14) | | |
| | *Sig.(p-value)* | *0.001 * * | *0.001 * * | *0.001 * * | | |
| RR | *Mean* | −67.259 | 75.994 | −147.003 | 0.470 | |
| | *SD* | 8.33 | 19.87 | 32.34 | 0.09 | 0.880 (*p = 0.001*) ** |
| | *Minimum* | −82.77 | 31.00 | −203.38 | 0.29 | |
| | *Maximum* | −45.40 | 108.98 | −84.99 | 0.67 | |
| RR-CL | *Mean difference (SE)* | −27.85 (±1.89) | −77.51 (±4.34) | −66.08 (±6.81) | | |
| | *Sig. (p-value)* | *0.001 * * | *0.001 * * | *0.001 * * | | |
| CL | *Mean* | −39.406 | 143.500 | −80.9206 | 0.501 | |
| | *SD* | 7.00 | 18.05 | 22.04 | 0.09 | 0.882 (*p = 0.001*) ** |
| | *Minimum* | −53.94 | 116.07 | −119.76 | 0.33 | |
| | *Maximum* | −29.09 | 180.06 | −42.88 | 0.68 | |
| CL-ML | *Mean difference (SE)* | −3.27(±1.89) | −8.44(±4.69) | −24.85(±4.46) | | |
| | *Sig. (p-value)* | *0.383* | *0.383* | *0.001 * * | | |
| ML | *Mean* | −36.130 | 161.939 | −56.071 | 0.644 | |
| | *SD* | 7.77 | 20.01 | 13.02 | 0.13 | 0.906 (*p = 0.001*) ** |
| | *Minimum* | −52.49 | 119.80 | −76.59 | 0.39 | |
| | *Maximum* | −26.80 | 185.97 | −31.83 | 0.84 | |
| Bone | *Mean* | −16.571 | 212.315 | −26.482 | 0.659 | |
| | *SD* | 4.35 | 11.20 | 8.25 | 0.11 | 0.709 (*p = 0.005*) ** |
| | *Minimum* | −25.55 | 189.19 | −40.01 | 0.44 | |
| | *Maximum* | −7.02 | 236.92 | −10.43 | 0.85 | |
| Bone-CL | *Mean difference (SE)* | 22.83 (±1.43) | 58.82 (±3.70) | 54.44 (±4.10) | | |
| | *Sig. (p-value)* | *0.001 * * | *0.001 * * | *0.001 * * | | |

* Statistical significance (Sig.) with Bonferroni correction $p \leq 0.01$; ** Sig. set at *p = 0.05*. Standard deviation (SD) and standard error (SE) have been reported. Acronyms in alphabetic order: ATT = attenuation; CL = connective; ELs = echo levels; EP = epithelium; GL = greyscale levels; ICC = intraclass correlation coefficient; ML = muscular layer; RR = rete ridges; TD = target depth.

For the quantitative ultrasound parameters, the assumption of homogeneity of variances was violated, as assessed by Levene's test for equality of variances ($p = 0.001$), so a Games–Howell test was performed to assess the differences between groups.

A comparison of ELs shows a difference between subgroups ($p = 0.001$). In particular, as regards the mucous layer, EP was not different from CL ($p = 0.831$), but these two tissues

are always separated by RR, which was found to have significantly lower EL than both EP ($p$ = 0.001) and submucosa (CL; $p$ = 0.001).

In the MM, the bone had mean ELs of −16,571 (±4.3) dB. The difference from the overlying CL was 22.83 (±1.43)—$p$ = 0.001. However, there appeared to be no significant difference between CL and ML in terms of EL ($p$ = 0.383). In BM and tongue, in terms of greyscale intensity, EP (149,205 ± 15.04), CL (143.5 ± 18.05), and ML (149,205 ± 15.04) appeared hypoisoechoic; there were no significant differences between them (Table 2). RR appeared hypoanechoic (75,994 ± 19.87—$p$ = 0.001), and bone appeared hyperechoic (212,315 ± 11.20—$p$ = 0.001), as shown in Figure 2b,d.

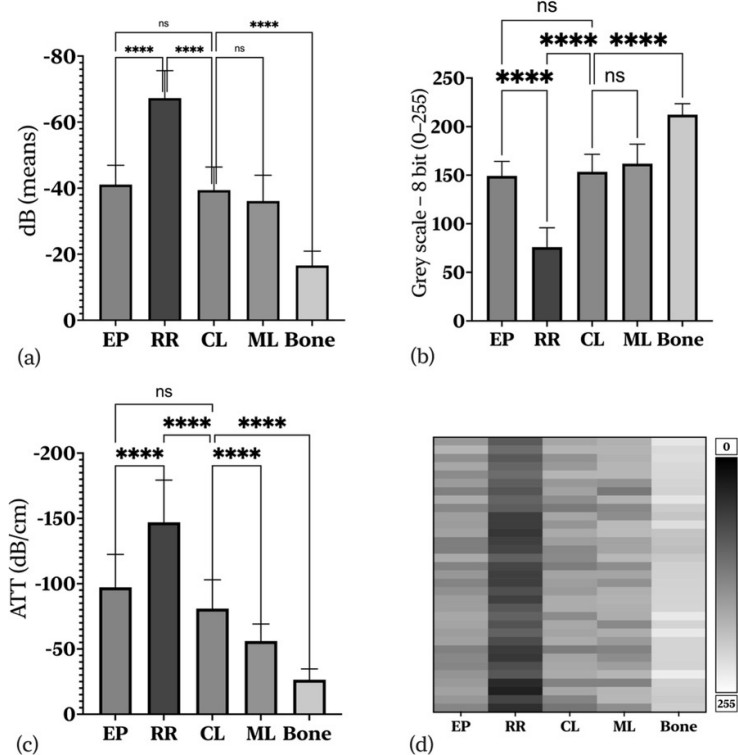

**Figure 2.** Histograms and heatmap showing the comparisons between groups; 95% C.I. is represented; ns = not significant; asterisk = the mean difference is statistically significant ($p \leq 0.01$). (**a**) Echo intensity comparison between groups. (**b**) Greyscale analysis between groups. (**c**) Relative attenuation (mean difference) between groups. (**d**) The heatmap shows the average of the ultrasound intensity variations, in grayscale, among the tissue layers in each analyzed case.

As for the ATT values, all the values are different from each other and specific for each tissue ($p$ = 0.001) (Table 2 and Figure 2c).

## 4. Discussion

The primary concern in evaluating the precision of an imaging technique is the level to which that procedure corresponds directly with histopathology. The ultrasonic images differentiate the tumor from the surrounding tissues in oral squamous cell carcinoma. Furthermore, the tumor's thickness is strongly correlated with the values obtained with ultrasound scanning and histological sections [20]. Magnetic resonance (MRI) and computed tomography (CT) represent the diagnostic imaging techniques of choice. However, these are expensive, and in the case of CT, the patient is exposed to a dose of radiation. Moreover, ultrasonography would have greater dimensional accuracy for oral squamous cell carcinoma at early stages with a thickness <5 mm [28].

A meta-analysis of studies correlating US tumor thickness (TT) measurements to histopathology concluded that US measures are highly accurate and only slightly over-

estimate TT values [29]. Another review reaffirmed the relationship of US and TT with histopathologic TT. Further, it examined the potential utility of different US parameters in oral tongue cancer management and the use of US as an intraoperative adjunct to resection [30].

The high diagnostic accuracy of US could also be helpful in the diagnosis of hyperplastic pathologies or other non-neoplastic lesions [31].

The major technical limitation of intraoral US is when large or posterior lesions are present, mainly due to difficulty in accessing the posterior tongue with a probe and the induction of vomiting [32]. However, based on the experience of the authors of this work, a probe with a T shape or hockey stick shape has ergonomic advantages, such as being able to cover most areas of the oral cavity, especially when compared with a classic linear one that does not have any angle but has the same handle thickness and scanning area. As an intrinsic limit of this technique, intraoral US is a strongly operator-dependent live examination: consequently, different radiologists with variable experiences may register different information [33].

To our knowledge, this is the first work to analyze the oral mucosa with ultrasonography from a quantitative point of view. For this exploratory study, the healthy mucosa was studied to lay the foundations for subsequent radiomics studies involving groups of patients with oral mucosal diseases. This type of study is already advanced in other areas of medicine, such as thyroid cancer, skeletal muscle pathologies, and breast cancer [24,25,34,35]. In the latter area, artificial intelligence systems are already present that recognize and classify lesions and have good approximations in terms of sensitivity and specificity [24]. The advantages of the clinical application of the quantitative analysis of ultrasounds and attenuation values are especially in ultrasound diagnostic improvement, which is still strongly operator-dependent [33].

Other studies have applied ultrasonography to analyze tissue interfaces in healthy mucosa and oral pathologies [18,23]. These studies laid the basis for defining tissue layers based on their echogenicity. Some have attributed grayscale values to the images, such as Izzetti et al. [36]. These authors used a high-frequency ultrasound (70 Mhz—HFUS) to define the tissue layers of the healthy mucosa. They assigned to each pixel a variable value encoded with a number between 0 and 256 using Horos software, which automatically samples the distribution of the gray level within a selected region of interest (ROI). They placed a fixed-area ROI at the mucosa–submucosa interface and retrieved data on the gray level distribution model (mean, standard deviation (SD), minimum, and maximum). As assessed for thickness values, mean echogenicity for each site was reported. Next, they obtained three measurements from each subject, thereby obtaining an average value for each subject at each site. The final value reported was the mean echogenicity.

The authors also described a gray value distribution for each anatomical site, attributing a range to the entire analyzed image (e.g., buccal mucosa or gingiva) without a quantitative analysis of each layer. With the same approach, they tried to identify various pathologies of the oral mucosa and anatomical alterations based on echogenicity [30]. Similar work had been performed on the oral mucosa using the optical coherence tomography (OCT) method [37–39]. Although pioneering, the limitations of this type of approach are that each tissue has its echogenicity; there are significant differences between one tissue and another. In the data presented in this study, one passes from a very hyperechoic tissue, bone (212.25 ± 3.70 GL) to a hypoanechoic tissue, RR (75.994 ± 19.87 GL). This last layer appeared always present and represented the transition from the mucousal to the submucosal layer. For this reason, defining it from a quantitative point of view could prove crucial for the qualitative and dimensional analysis (tumor thickness, depth of invasion, inflammatory and autoimmune diseases analysis) of the pathologies of the oral mucosa.

It was also possible to define each tissue identified based on specific physical characteristics of the insonated tissues (impedance, attenuation, and backscatter) by reporting the "raw" values in dB of each identified tissue. This tissue definition adds more information than gray analysis because it translates quantitative ultrasound information into qualitative

information. The characteristic echogenicity of the RR could be due to the interaction of the acoustic wave with any inhomogeneity in a tissue characterized by the connective epithelial interface [27,40–42].

Finally, it was possible to calculate each tissue's attenuation values (ATT-dB/cm), correcting any distortion given by the impedance and the distance from the frequency emission. While the difference in dB between ML and CL was not significant at the EL ($p = 0.838$), the two layers were different ($p = 0.001$) when normalized in two-dimensional space. These data could be significant in building a quantitative analysis model based on artificial intelligence (AI).

This study had two main limitations. First, an ultrasound system allows the acquisition of many parameters, even with standardized presets; in any case, these parameters are specific for each device. In order to generalize the computerized analysis in real time, the physical characteristics of each ultrasound equipment should be tested through a phantom. In this way, it could be possible to establish the contrast parameters for a specific spatial configuration [24,27]. The latter concerns the definition of ROIs. Since there are different fabrics with different thicknesses, it is not always possible to standardize the area of interest with the possibility of altering some data. The ML, for example, is very inhomogeneous, since it is characterized by the alternation of the iso-hyperechoic regions and anechoic areas, representing the muscle striae from an ultrasound point of view (Figure 1). The ROI should encompass both areas, but the resulting mean values may not accurately define the tissue. Further studies with more data are needed to determine the feasibility of a standardized quantitative ultrasound image analysis.

## 5. Conclusions

Quantitative ultrasound analysis of the oral mucosa allows distinguishing the tissues in a specific way, not only based on the visible echogenicity in grayscale, but also through the attribution of ultrasound parameters characteristic of each tissue (EL and ATT). This type of image analysis could hold promise in applications for oral mucosal pathologies and serve as a basis for developing radiomics models and computerized image analysis. This type of development could help decrease operator-dependent diagnostic variability and allow for the use of this technique by oral medicine specialists.

**Author Contributions:** Conceptualization, D.D.S. and A.L.; investigation, A.R. and M.M.; resources, D.D.S., M.C., M.P. and I.H.; data curation, D.D.S. and A.L.; writing—original draft preparation, D.D.S.; writing—review and editing, A.L.; visualization, R.S. All authors have read and agreed to the published version of the manuscript.

**Funding:** This research received no external funding.

**Institutional Review Board Statement:** The Biomedical Institutional Review Board of the University of Campania Luigi Vanvitelli, Naples, Italy (#339/2018), has approved this prospective descriptive study, wherein patients were enrolled following the Declaration of Helsinki and its later amendments.

**Informed Consent Statement:** Informed consent was obtained from all subjects involved in the study.

**Data Availability Statement:** Data sharing not applicable.

**Acknowledgments:** The authors would like to acknowledge all the patients who participated in the study.

**Conflicts of Interest:** The authors declare no conflict of interest.

## Abbreviations

| | |
|---|---|
| A-mode | amplitude mode |
| ATT | attenuation |
| B-mode | brightness mode |
| BM | buccal mucosa |

| CT | computed tomography |
|---|---|
| CL | connective tissue |
| dB | decibels |
| EL | echo level |
| EP | Epithelium |
| GL | greyscale level |
| HFUS | high-frequency ultrasound |
| ICC | intraclass correlation coefficient |
| MRI | magnetic resonance |
| MM | masticatory mucosa |
| ML | muscular layer |
| NBI | narrow-band imaging |
| OCT | optical coherence tomography |
| RCM | reflectance confocal microscopy |
| ROI | region of interest |
| RR | rete ridges |
| TD | target depth |
| TT | tumor thickness |
| US | ultrasonography |

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
