# Peer review of "Quantitative Ultrasound Analysis of Oral Mucosa: An Observational Cross-Sectional Study"

_applsci, doi:10.3390/app12146829_

Round 1

Reviewer 1 Report

Required lot of efforts obviusly but it is not clear is it for radiologists or for oral pathologists. It is also not clear how to apply US on all regions of oral cavity. As it is already said this can be begining of something...

Author Response

  1. Required lot of efforts obviusly but it is not clear is it for radiologists or for oral pathologists.
    1. Thanks for the suggestion. The phrase “This type of development could help decrease operator-dependent diagnostic variability and allow for the extension of this technique to oral medicine specialists” has been added into the main text in the Conclusion section.
  2. It is also not clear how to apply US on all regions of oral cavity. As it is already said this can be begining of something..
    1. Thanks for the valuable suggestion. The complete regions and areas scanned in this study have been reported in a new Table 1. Moreover, the phrase “However, based on the experience of the authors of this work, a probe with a T-shape or hockey stick has an ergonomics such as being able to cover most areas of the oral cavity, especially when compared with a classic linear one that does not have any angle but the same thickness of handle and scanning area” has been added in the Discussion section among the technical limitations (lines 245-251).

Reviewer 2 Report

Dear Authors,

The article: 'Quantitative ultrasound analysis of oral mucosa: an observational cohort study' was to perform a quantitative analysis of healthy oral mucosa based on the analysis of greyscale (GL), echo levels (EL - dB), and attenuation values (ATT - dB/cm). 

English language and style must be corrected.

Numerous punctuation mistakes should be corrected. 

The introduction is well written. 

Materials and methods

The p value should be written in italic

Discussion is clearly presented.

Add table with abbeviations before references.

References should be prepared according MDPI guidelines.

 Article can be accepted after major revision.

Author Response

Dear Authors,

The article: 'Quantitative ultrasound analysis of oral mucosa: an observational cohort study' was to perform a quantitative analysis of healthy oral mucosa based on the analysis of greyscale (GL), echo levels (EL - dB), and attenuation values (ATT - dB/cm). 

  1. English language and style must be corrected.
    1. Thanks for the suggestion. English language and style have been corrected by one of the co-authors who’s a native English speaker
  2. Numerous punctuation mistakes should be corrected. 
    1. The punctuation has been corrected through software analysis
  1. The introduction is well written. Materials and methods. The p value should be written in italic. Discussion is clearly presented.
    1. We thank the reviewer. All the p-values are converted in italic.

  1. Add table with abbreviations before references.
    1. A section named Abbreviations has been added to the main text before the references
  2. References should be prepared according to MDPI guidelines.
    1. References have been upgraded using the Mendeley Reference manager software.

Article can be accepted after major revision.

Reviewer 3 Report

Interesting paper but in need of significant revision, there are any grammar issues and typos, revise

There are many acronyms, such as 'US" which can be confusing, just say'Ultrasonography', the same for table 2, figures 1 and 2 footnotes, spell out or add foot notes

The location of the scanned areas is not precise, add a table providing this detail, 

One of the possible usages of Ultrasonography is for exposing impacted tooth/teeth, and to establish if the tooth is covered by bone or not, as this can determine using less invasive methods such as diode laser or the conventional scalpel exposure, expand on this (Compend Contin Educ Dent. 2017 Apr;38(eBook 5):e18-e31.; Photonics. 2022; 9(4):265. https://doi.org/10.3390/photonics9040265)

Figure 1, not clear which area of the mouth this view is? expland

very deficient introduction and discussion, revise

Author Response

1. Interesting paper but in need of significant revision, there are any grammar issues and typos, revise.  

a. Thanks for the suggestions. Grammar issues and typos have been revised. In particular, the English form has been checked by one of the co-authors who’s a native-English speaker.

2. There are many acronyms, such as 'US" which can be confusing, just say'Ultrasonography', the same for table 2, figures 1 and 2 footnotes, spell out or add footnotes

a. Thanks for the valuable suggestion. Regarding the acronyms, a section called “Abbreviations” has been added in the main text before the references; the footnotes of table 2 have been modified. Moreover, in the caption of Figure 1, the acronyms have been written in the extended form (e.g. EP epithelium).

3. The location of the scanned areas is not precise, add a table providing this detail.

a. A table with the scanned areas has been added and named Table 1;

b. Moreover, the phrase “However, based on the experience of the authors of this work, a probe with a T-shape or hockey stick has an ergonomics such as being able to cover most areas of the oral cavity, especially when compared with a classic linear one that does not have any angle but the same thickness of handle and scanning area” has been added to the Discussion section.

4. One of the possible usages of Ultrasonography is for exposing impacted tooth/teeth, and to establish if the tooth is covered by bone or not, as this can determine using less invasive methods such as diode laser or the conventional scalpel exposure, expand on this (Compend Contin Educ Dent. 2017 Apr;38(eBook 5):e18-e31.; Photonics. 2022; 9(4):265. https://doi.org/10.3390/photonics9040265)

a. Thanks for the important suggestion. The phrase “One of the possible usages of Ultrasonography is for exposing impacted tooth/teeth, and to establish if the tooth is covered by bone or not, as this can determine using less invasive methods such as diode laser or the conventional scalpel exposure” has been added to the main text in the Introduction section.

5. Figure 1, not clear which area of the mouth this view is? expland

a. Figure 1 caption has been modified: Two examples of ROI sampling in two anatomical regions of the oral cavity: (a) dorsal tongue; 1. epithelium, 2. rete ridges, 3. connective layer, 4. muscle layer. (b) ultrasound imaging of the upper gingiva and alveolar bone (masticatory mucosa). 1. epithelium, 2. rete ridges, 3. connective layer, 4. alveolar bone

6. very deficient introduction and discussion, revise

a. Thanks for the suggestion. The introduction and Discussion sections have been improved following the reviewer’s comments.

Reviewer 4 Report

Thanks for this interesting paper.

I would like to congratulate the authors for this nice build-up of the paper. The introduction is satisfactory which justified the onset of this research work (this observational study). But I would like to raise the attention of the authors that this is not a cohort study since no follow-up was conducted for the recruited 30 patients.  Therefore, it is an observational cross-sectional study. Please correct this in the title of the paper as well as in the relevant sections.

This will lead us to the second point which is the sampling procedures. I would be enthusiastic to know the sampling method used to recruit these 30 patients. What kind of sampling did you use? What was your sampling frame?

In relation to the inclusion criteria: did you accept patients undergoing orthodontic treatment? or patients with severe broken-down teeth? Did you accept patients with cleft lip and palate? Did you accept patients with previous trauma to the maxillofacial complex? Please strengthen your inclusion criteria.

In the 'Equipment' section, we need please more information about the devices used with their manufacturers, cities, and countries. Some information is missing in this section of the manuscript.

Table 1 is not really important. Please consider deleting it. The information about the parameters used in the scanning stage by this apparatus can be given within the paragraph.

All the products, programs, and tools used in this study should be mentioned along with the companies' names, their cities, and countries. The authors ignored giving full information in different areas of the manuscript.

The anatomic regions that were scanned are not given in the manuscript (in the Materials and Methods section). Please mention all the areas that had been scanned and analyzed.

Otherwise, the rest of the paper is fine. The discussion has enough depth and the citations are up-to-date.

Author Response

Thanks for this interesting paper.

- I would like to congratulate the authors for this nice build-up of the paper. The introduction is satisfactory which justified the onset of this research work (this observational study). But I would like to raise the attention of the authors that this is not a cohort study since no follow-up was conducted for the recruited 30 patients. Therefore, it is an observational cross-sectional study. Please correct this in the title of the paper as well as in the relevant sections.

We would like to thank the reviewer. As suggested the study design has been corrected.

- This will lead us to the second point which is the sampling procedures. I would be enthusiastic to know the sampling method used to recruit these 30 patients. What kind of sampling did you use? What was your sampling frame?

Thanks for the crucial suggestion. The Materials and Methods section has been modified: Patients were recruited consecutively at the Oral Pathology Unit at the University of Campania L. Vanvitelli, Naples, Italy, from December 2021 to March 2022 after giving written informed consent. The final number of patients enrolled in the study (who met the inclusion and exclusion criteria) has been left in the Result section: A total of 33 patients (17 females and 16 males; mean age 58,42±13.29 y.o; age range 29–80 years) were considered for this study.

- In relation to the inclusion criteria: did you accept patients undergoing orthodontic treatment? or patients with severe broken-down teeth? Did you accept patients with cleft lip and palate? Did you accept patients with previous trauma to the maxillofacial complex? Please strengthen your inclusion criteria.

Thanks for the precious suggestions. The Materials and methods section has been modified and the inclusion/exclusion criteria have been improved: Patients over the age of 18 with at least one or more sites of healthy oral mucosa were included in the study. The following lesions and or conditions were considered as exclusion criteria: a) oral potentially malignant disorders; b) oral squamous cell carcinoma; oral pigmented lesions; c)erosive-erythematous lesions of inflammatory and/or auto-immune etiology; d) severe broken-down teeth; e) cleft lip; f) palate lip; g) orthodontic treatments; h) smoking and alcohol consumption; h) pregnant patients

- In the 'Equipment' section, we need please more information about the devices used with their manufacturers, cities, and countries. Some information is missing in this section of the manuscript.

The equipment section has been modified following the instruction of the reviewer. Moreover, all the information on the tools (hardware or software) used in this study has been upgraded.

- Table 1 is not really important. Please consider deleting it. The information about the parameters used in the scanning stage by this apparatus can be given within the paragraph.

Thanks for the suggestion. The former table 1 has been deleted and the information has been transferred in the main text: Images have been acquired in B-mode at the maximum transmit frequency (18 MHz, harmonic), a gain set on 45, and 60dB of dynamic range. The examination was performed after applying ultrasound gel to the scanning area. A fixed distance of about 5 mm between the area to scan and the probe has been set. The focus has been standardized

- All the products, programs, and tools used in this study should be mentioned along with the companies' names, their cities, and countries. The authors ignored giving full information in different areas of the manuscript.

Thanks for the suggestion. All the products, programs, and tools used in this study have been mentioned along with the companies' names, their cities, and countries

- The anatomic regions that were scanned are not given in the manuscript (in the Materials and Methods section). Please mention all the areas that had been scanned and analyzed. Otherwise, the rest of the paper is fine. The discussion has enough depth and the citations are up-to-date.

Thanks for the suggestion. The complete regions and areas scanned in this study have been reported in a new Table 1.

Round 2

Reviewer 2 Report

article can be accepted after Editor decision

Author Response

We want to thank Reviewer 2.

Reviewer 3 Report

Thank you for the revision

Abstract, don't use acronyms in the abstract, you have enough space

revise 'Regions of interest 26 (ROI) have been traced for the analysis of echo levels - EL (in dB) grey levels (GL) and attenuation 27 values (ATT dB/cm).'

Methodology

add a subheading for the 'study sample', and add these sentences from the results 'A total of 33 patients (17 females and 16 males; mean age 58,42±13.29 y.o; age range 2980 197 years) were considered. for this study. The fullentire series encloses a total of 1002 ultra-sound images of the oral cavity covering all the anatomy sites: 334 images of masticatory mucosa (MM),, 328 images of buccal mucosa (BM),, and 340 images of the tongue. Ten  images that met the quality standards were selected for each patient, for a total of 330 im- 201 ages of healthy mucosa analyzed. In each image, it was possible to identify all the tissue 202 layers based on the anatomical site.' 

change the '2.1 Equipment' to '2.1 ultra-sound investigations' also add the city, and country, for the GE Logiq-e R7 right after it

Results, add a footnote for acronyms used in the table

was there any significant difference between bone and tooth tissue?

Author Response

Thank you for the revision

  1. Abstract, don't use acronyms in the abstract, you have enough space

revise 'Regions of interest 26 (ROI) have been traced for the analysis of echo levels - EL (in dB) grey levels (GL) and attenuation 27 values (ATT – dB/cm).

Thanks for the suggestion. The Abstract section has been modified.

  1. Methodology: add a subheading for the 'study sample', and add these sentences from the results 'A total of 33 patients (17 females and 16 males; mean age 58,42±13.29 y.o; age range 29–80 197 years) were considered. for this study. The fullentire series encloses a total of 1002 ultra-sound images of the oral cavity covering all the anatomy sites: 334 images of masticatory mucosa (MM),, 328 images of buccal mucosa (BM),, and 340 images of the tongue. Ten  images that met the quality standards were selected for each patient, for a total of 330 im- 201 ages of healthy mucosa analyzed. In each image, it was possible to identify all the tissue 202 layers based on the anatomical site.' 

Thanks. The Materials and methods section has been modified following the reviewer's suggestions and adding the 2.2 Study sample subsection. The numbering of the subsequent sub-paragraphs has been adjusted accordingly. Moreover, the phrase above has been removed from the Results section.

  1. change the '2.1 Equipment' to '2.1 ultra-sound investigations' also add the city, and country, for the GE Logiq-e R7 right after it

Thanks. The Equipment subheading has been changed into “2.1 Ultrasound investigations”. Information about the manufacturer moved from the end of the period to the right after GE Logiq-e R7.

  1. Results, add a footnote for acronyms used in the table.

Thanks for the suggestion. Table 2 footnotes have been modified, and acronyms in alphabetic order have been added.

  1. was there any significant difference between bone and tooth tissue?

  Thanks for the question. The study focuses on the evaluation of oral mucosa. The teeth were not analyzed. This tip will be helpful for future studies on the subject.

Round 3

Reviewer 3 Report

Thank you for the revision